# Optimizing Fine Motor Coordination, Selective Attention and Reaction Time in Children: Effect of Combined Accuracy Exercises and Visual Art Activities

**DOI:** 10.3390/children10050786

**Published:** 2023-04-27

**Authors:** Mohamed Frikha, Raghad Saad Alharbi

**Affiliations:** 1Department of Physical Education, College of Education, King Faisal University, Al-Ahsa 31982, Saudi Arabia; 220001448@student.kfu.edu.sa; 2Research Laboratory-Education, Motricity, Sport and Health (LR19JS01), High Institute of Sport and Physical Education, Sfax University, Sfax 3000, Tunisia

**Keywords:** motor coordination, motor accuracy, motor learning, reaction time, selective attention, ceramics, pottery, raw materials, middle childhood

## Abstract

Optimizing fine motor coordination and cognitive abilities in middle childhood through new intervention programs remains one of the most essential orientations in motor development and skills proficiency. The present study aims to identify the effect of a structure-based program intervention, combining motor accuracy exercises with visual art activities (ceramics, pottery, and creations using raw materials), on fine motor coordination, selective attention, and reaction time in middle childhood. Sixty, right-handed typically developed male schoolchildren (age = 8.29 ± 0.74 years; body height = 130.25 ± 0.05 cm and body mass = 29.83 ± 4.68 kg; mean ± SD) volunteered to participate in the study. They were randomly assigned to a combined group intervention (MG) receiving a mixed program integrating visual art activities and motor accuracy exercises; an accuracy group intervention (AG) receiving oriented motor accuracy exercises; and a control group (CG). Selective attention, reaction time, and fine motor coordination skills (accuracy: darts throw; manual dexterity: grooved pegboard test) were measured. Repeated measure ANOVA, one-way ANOVA, and Kruskal–Wallis ANOVA were performed for analysis. The results showed higher changes in MG compared to AG in manual dexterity (*p* < 0.001), in motor accuracy (*p* < 0.001), and in reaction time (*p* < 0.01), but not in selective attention (*p* = 0.379). In addition, higher changes were recorded in AG compared to CG in motor accuracy (*p* < 0.05), reaction time (*p* < 0.05), and in selective attention (*p* < 0.01), but not in manual dexterity (*p* = 0.082). The structured combined program best assists accuracy, manual dexterity, reaction time, and selective attention. Therefore, instructors in sports academies and teachers at schools are advised to use the combined program in the training sessions and in the non-curricular activities, respectively, to improve fine motor coordination, selective attention, and reaction time in middle childhood.

## 1. Introduction

The interest in fine motor skills and their improvement in children is still growing [1,2]. Children spend between 30% and 60% of their school time performing fine motor tasks in which pencil-based activities take up nearly 85% of the spent time [3]. Therefore, the need for innovative methods and training programs, aiming for the improvement of motor skills in children, is still rising [4], particularly with the observed sedentary behaviors [5,6], and the decrease motor competency in children [7,8]. Indeed, physical inactivity (i.e., low level of PA) induces the emergence of health risk factors [9] and reduces motor competency [10,11]. Several phenomena make the situation more serious. Indeed, previous studies demonstrated that motor skills are negatively affected by the excessive use of technology, and screen time to which children are exposed, which were found to limit their motor experiences and affect their quality of life [12,13]. Moreover, recent studies revealed a decline in gross and fine motor skills after the COVID-19 pandemic restriction period [14,15]. In contrast, regular PA in childhood is indispensable in both gross and fine motor skills development [10], where it promotes the improvement of fundamental motor skills and supports cognitive development over the long term [4].

Promoting motor skills in children through PA was confirmed to substantially and positively affect physical, mental, emotional, and cognitive development, with a more noticeable effect of open-ended, strategic, and sequential motor skills on cognitive abilities [16]. Several research studies have demonstrated the relationship between motor skills acquisition and cognitive abilities. As a matter of fact, specific motor tasks can affect cognitive abilities such as attention [17], executive function [18], working memory [19], and cognitive flexibility differently [20]. Furthermore, it was confirmed that structured PA, performed under the guidance and direct instruction of adults or instructors, has been assumed to be more helpful for motor skills development than non-formal sports activities (i.e., unstructured PA) [4]. 

The scientific literature shows an equivalent effect of visual arts activities on cognitive abilities and fine motor skills. Indeed, Ghasemi et al. [21] reported that pottery learning had a significant effect on improving participants’ creativity, and their motor-cognitive skills. Samadzadeh et al. [22] demonstrated that visual arts are effective in the education of coping strategies in annoyed children. Hina [23] mentioned that rehabilitation through art activities improves cognitive function, and conceptual and perceptual skills in children aged 4–9 years, while Erim and Caferoğlu [24] highlighted the importance of the inclusion of visual art activities in the curriculum for motor skills improvement in mentally handicapped children. Recently, Bradeško and Potočnik [25] found that the use of art activities had a positive effect on perseverance, attention, and concentration in students with deficits in individual learning areas. 

In general, either visual art or oriented PA contributed to the enhancement of fine motor skills and cognitive abilities. To the authors’ best knowledge, no previous research had investigated the effect of their combination on the promotion of both fine motor coordination and cognitive abilities in children. Therefore, the present study aimed to investigate the effect of structure-based physical activity interventions, using combined motor accuracy exercises and visual art activities, on fine motor coordination, selective attention, and reaction time in male schoolchildren aged 7–9. We hypothesized that combining oriented motor accuracy exercises and visual art activities (the fields of shaping with ceramics, shaping with materials) during structured PA best assists fine motor coordination (accuracy, manual dexterity), selective attention, and reaction time in typically developing male schoolchildren of 7–9 years of age.

## 2. Materials and Methods

### 2.1. Participants

Sixty, right-handed male schoolchildren (age = 8.29 ± 0.74 years; body height = 130.25 ± 0.05 cm and body mass = 29.83 ± 4.68 kg; mean ± SD), regularly attending school, not suffering from mental or motor disabilities and not taking any medication, volunteered to participate in the study. A priori sample size was determined using G*Power software (3.1.9.2), given an effect size = 0.25 (Medium), α = 0.05, power 1–β = 0.95, and was set at 57 participants. Written informed consent was obtained from participants’ parents after receiving a thorough explanation of the protocol, the benefits, and the risks involved. The participants were primary school students receiving two hours of physical education sessions per week according to the school curriculum, declared as having no previous experience in the testing tasks, and were all affiliated with Al-Madinah Sports Academy (Al-Madinah Al-Munawara, Saudi Arabia). The inclusion criteria were (i) being affiliated with the academy; (ii) not suffering from any intellectual or physical disabilities; (iii) having a rate of attendance in the proposed program > 75%; and (iv) completing all testing sessions. Three participants were removed from the study (two from the AG and one from the CG), for missed training sessions (nonattendance > 25%) and for missed testing sessions, respectively. Hence, the data from 57 participants were retained for analysis. The protocol was conducted in accordance with the Declaration of Helsinki (1975, revised 1983) and approved by King Faisal University Ethics Committee (Ethical Clearance KFU-REC-2021-OCT-EA00077).

### 2.2. Procedures

During the week preceding the structured PA programs, participants familiarized themselves with the testing procedures in two separate sessions. These familiarizations ensured that participants were fully knowledgeable of the experimental conditions and measurement procedures. Then, they were randomly assigned to either a first group intervention receiving a combined program (mixed group; MG) integrating visual art activities and motor accuracy exercises; a second group intervention (motor accuracy group; AG) receiving a training program oriented to motor accuracy; and a third group designed as a control group (CG).

The proposed structured PA programs and the assessment procedures were conducted by the same experimenter at the same habitual session time in the academies (i.e., 5:00–7:00 p.m.). Data from familiarization sessions were retained for rest-retest reliability. All test-retest reliability coefficients were acceptable as were set between ICC = 0.752 and ICC = 0.813.

### 2.3. Structure-Based Program Interventions 

All programs’ durations lasted 8 weeks with 24 learning-training sessions (evaluation sessions included), with three training sessions per week. The duration of each session was 60 min. All sessions took place in the sports academies outside of school time (5:00 to 7:00 p.m.). Participants had to follow the instructions of the experimenter while executing the proposed exercises. All intervention programs were implemented in the period from 11 December 2021 to 16 March 2022.

#### 2.3.1. The Motor Accuracy Program (AG)

A differentiated approach was adopted in which the task difficulties were altered (varying throw distances, varying throw targets, combining the target sizes to throw distances, and using different sizes, weights, and shapes of the thrown items) according to Ong et al. [26] and Gaspar et al. [27]. The motor accuracy program included motivating games and competitions. Participants were instructed to throw different objects in unconventional ways to increase their individual ability to adapt to the new movement patterns. A standard warm-up of 10 min using static stretching exercises was performed at the beginning of each session according to the protocol of Frikha et al. [28]. In this intervention protocol, the experimenter provided correctional verbal and visual feedback to participants when he thought that this was necessary. In general, each learning-training session included two exercises and one game (Table 1). 

#### 2.3.2. The Mixed Visual Art and Motor Accuracy Program (MG)

The mixed program included visual arts activities and motor accuracy exercises which were altered and integrated into the same learning-training program. The visual art activities included ceramic, pottery formation, and creations using raw materials (Table 2), while the motor accuracy part included exercises and games from the motor accuracy program. Each training session included 35 min of art activities and 15 min of oriented motor accuracy games, competitions, and exercises, conducted with variations of instructions and difficulty executions. 

### 2.4. Measures

#### 2.4.1. The Belt Test (i.e., a Paper-Pencil Test; BT)

The belt test (BT) is a psychometric task, which measures visual-spatial ability, as well as selective attention. Selective attention refers to the processes that enable an individual to pick and focus on certain inputs for subsequent processing while ignoring irrelevant or distracting information [29]. The subject must circle all the targets (bells) that are encountered. Targets (35 bells) are distributed equally in seven columns. In each column, there is the same number of targets (N = 5) and distracters (N = 40). All drawings are black–like Chinese shadows. Subject performance is evaluated quantitatively (number of bells crossed and omitted). The duration of the test is 3 min, and a higher number of correct responses reflects a better performance [30].

#### 2.4.2. The Reaction Time (Green Light Reaction Time Test; RT) 

Reaction time is defined as the s interval of time between the presentation of a stimulus and the appearance of an appropriate voluntary response in a subject [31]. A simple reaction time test is assumed to be a valid measure of cognitive function in healthy subjects [32]. An online reaction time test was used (https://faculty.washington.edu/chudler/java/redgreen.html, last accessed on 16 March 2022). Participants had to click the spacebar on the keyboard as fast as the green stoplight appeared. A laptop (HP, Intel core i7^®^) calculated the RT (in ms). Ten trials were given to each participant. Higher scores reflect a poorer performance. 

#### 2.4.3. Darts Throwing Test (DT)

This task aims to assess accuracy and hand-eye coordination. Facing a dartboard (45.72 cm diameter with a series of 10 concentric rings) hung on a wall at 2.37 m away (with the possible regulation of height according to the eye level of each participant); the participant was instructed to perform 10 throws. Each throw was scored according to the position of the dart on the board (from zero to 10). A dart that missed the board or that bounced off was given a score of “zero” [33]. Dart throwing accuracy and consistency were evaluated using three calculations: first the total score of the ten throws. This score could range from zero (all misses) to 100 (all bullseyes); second, the number of errors (number of times the target was missed), which could range from zero to 10; and third, the variability of scores: VAR = [SD scores]/[mean score]. A lower VAR indicates a higher consistency [26].

#### 2.4.4. The Grooved Pegboard Test (GPT)

The grooved pegboard test (GPT) assesses fine manual dexterity, motor speed, and hand-eye coordination [32]. The apparatus (Lafayette instruments^®^ 32025, Lafayette, LA, USA) consists of a 10.1 cm by 10.1 cm metal surface with a 5 by 5 matrix of keyhole-shaped holes in varying orientations. Each peg is 3 mm in diameter with a small ridge running along its 2.5 cm length. The GPT consists of 25 holes with randomly positioned slots. Pegs, which have a key along one side, must be rotated to match the hole before they can be inserted. The GPT followed a procedure validated previously [34,35,36]. The pegboard is placed in mid-line with the subject so that the board is at the edge of the table and the peg tray is just above the board. Participants have to put the pegs into the boards as fast as possible using only one hand (dominant or non-dominant), fill the top row completely from side to side, and then the remaining rows the same way as they filled the top row. For the right-hand trial, the instructor demonstrates that the pegs are placed from the subject’s left to right, and from right to left for the left-hand trial. The dominant hand trial is administered first, followed by the non-dominant hand trial. Participants are explicitly instructed not to use their free hand for assistance.

#### 2.4.5. The Perceived Competence (PC) 

The self-perceived competence was assessed before task execution in the test and retest sessions, using one adapted item from Fredriks and Eccles [37]. Participants responded to a single question “How do you think you will perform on the follow-up task?”, on a 7-point Likert scale (1 = very poorly; 7 = very well). The internal consistency of the scale was set at (Cronbach α = 0.81), which is in concordance with Frikha et al. [38].

#### 2.4.6. The Perceived Difficulties (PD)

The self-perceived difficulty was assessed after task execution in the test and retest sessions using the DP15 scale [37]. Participants rated their perceived difficulty using a 15-point item questionnaire (1 = extremely easy; 15 = extremely difficult). The reliability of the scale was set at 0.859, which is in accordance with Delignière et al. [39]. 

### 2.5. Statistical Analysis

All statistical tests were processed using STATISTICA Software (Stat Soft, France). Data are reported as mean ± SD. Once the assumption of normality (Shapiro–Wilk’s W Test) was confirmed, parametric tests were used, and one-way ANOVA in addition to repeated measure ANOVA were performed. If not, non-parametric tests and Kruskal–Wallis ANOVA were performed. Significant differences between means were assessed using the Tukey HSD. Test-retest reliability was assessed using repeated-measures analysis of variance and intra-class correlation. Furthermore, the effect size “partial η^2^” for significant main effects was calculated. The thresholds included small (d < 0.2), medium (0.3 < d < 0.5), and large (0.5 < d) [40]. The power of statistical tests was verified with the G*Power software version (3.1.9.2). Considering the sample size, the significant level of 5%, and the partial effect size η^2^, the calculated power analyses (1–β) values were verified and were set between 0.8 and 1.00 for all variables. Statistical significance was set, a priori, at *p* < 0.05.

## 3. Results

### 3.1. Belt Test (BT) and Reaction Time (RT) Tests Performances

The results of BT and RT are presented in Figure 1. Concerning BT scores, the Kruskal–Wallis ANOVA showed significant differences in test sessions between MG vs. CG and AG vs. CG (*p* = 0.045 and *p* = 0.033, respectively). However, in the retest session, only a difference between MG vs. CG was observed (*p* < 0.01). The Wilcoxon matched pairs test showed that BT improved significantly from test to retest sessions in MG (*p* < 0.001) and AG (*p* < 0.01), but not in CG (*p* > 0.05).

Concerning changes in BT (ΔBT), the one-way ANOVA showed a significant group effect (F = 10.105; *p* < 0.001; η^2^ = 0.272, small). The Tuckey HSD showed that ΔBT was higher in MG compared to CG (*p* < 0.001) and in AG compared to CG (*p* < 0.01). However, no significant difference was recorded between MG and AG (*p* = 0.379).

In relation to changes in BT errors (ΔBTe), the one-way ANOVA showed a significant group effect (F = 9.967; *p* < 0.001; η^2^ = 0.269, small). The Tuckey HSD showed that ΔBTe was higher in MG compared to CG (*p* < 0.001) and in AG compared to CG (*p* < 0.01).

Concerning the reaction time (RT), the repeated measure ANOVA showed a significant measure effect (F = 46.685; *p* < 0.001; η^2^ = 0.463, medium) and a significant interaction measure × group (F = 16.459; *p* < 0.001; η^2^ = 0.378, medium). The Tuckey HSD test showed that MG and AG significantly improved their performances (*p* < 0.001 and *p* < 0.01, respectively).

In relation to changes in reaction time (ΔRT), the one-way ANOVA showed a significant group effect (F = 16.458; *p* < 0.001; η^2^ = 0.378, medium). The Tuckey HSD test showed that ΔRT was higher in MG compared to AG (*p* < 0.01) and in AG compared to CG (*p* < 0.05).

### 3.2. Darts Throwing Accuracy (DT) and Grooved Pegboard Tests (GPT)

The results of DT and GPT are presented in Figure 2. Concerning the DT, the repeated measure ANOVA showed a significant effect on the variable group (F = 15.716, *p* < 0.001, η^2^ = 0.367, medium). The interaction group × measure was significant too (F = 55.293, *p* < 0.001, η^2^ = 0.671, large). The post hoc analysis showed higher DT scores in MG and AG in retest compared to test scores (*p* < 0.001). No difference was detected between test and retest scores in CG (*p* = 0.949). In addition, a higher retest DT score was detected in MG compared to AG (*p* < 0.001) and in AG compared to CG (*p* < 0.001). Concerning the changes in DT, the one-way ANOVA showed higher changes in MG compared to AG (*p* < 0.001) and in AG compared to CG (*p* < 0.05).

Concerning the GPT, the repeated measure ANOVA showed an effect of the variable measure (F = 30.879, *p* < 0.001, η^2^ = 0.363, medium). However, no group effect was detected (F = 0.674, *p* = 0.513). The interaction group × measure was significant (F = 18.948, *p* < 0.001, η^2^ = 0.412, medium). The post hoc analysis showed higher values in the test compared to the retest session in the MG group (*p* < 0.001). However, no significant differences were detected between test and retest values in AG (*p* = 0.159) and in CG (*p* = 0.984). Concerning the changes in GPT, the one-way ANOVA showed higher changes in MG compared to AG and CG (*p* < 0.001).

In relation to the variability coefficient (VAR) of DT scores, the repeated measure ANOVA showed a significant group effect (F = 12.057, *p* < 0.001, η^2^ = 0.308, medium), a significant measure effect (F = 75.324, *p* < 0.001, η^2^ = 0.582, large) and a significant interaction group × measure (F = 14.423, *p* < 0.001, η^2^ = 0.348, medium). The post hoc analysis showed higher VAR in the test compared to retest values in both MG and AG (*p* < 0.001). However, no significant changes in VAR were detected in CG (*p* = 0.799). 

### 3.3. Perceived Competence and Perceived Difficulties

#### 3.3.1. The Grooved Pegboard Test

Concerning the self-perceived competence (PC), the repeated measure ANOVA showed significant group and measure effects (F = 6.183, *p* < 0.01, η^2^ = 0.186, small; F = 66.467, *p* < 0.001, η^2^ = 0.551, large, respectively). The interaction group × measure was significant too (F = 8.673, *p* < 0.001, η^2^ = 0.243, medium). The post hoc analysis showed higher PC in retest compared to test values in MG and AG (*p* < 0.001 and *p* < 0.01, respectively) and higher retest PC perceptions in MG and AG compared to CG (*p* = 0.003 and *p* = 0.006, respectively).

In relation to the self-perceived difficulty (PD), the repeated measure ANOVA showed a significant group and measure effects (F = 14.189, *p* < 0.001, η^2^ = 0.344, medium; F = 138.996, *p* < 0.001, η^2^ = 0.720, large, respectively). The interaction group × measure was significant too (F = 34.962, *p* < 0.001, η^2^ = 0.564, large). The post hoc analysis showed lower PD perceptions in retest compared to test values in MG and AG (*p* < 0.001 and *p* < 0.001, respectively) and lower retest PD in MG and AG compared to CG (*p* < 0.001 for both comparisons) (Table 1).

#### 3.3.2. The Darts Throw Test

Concerning the self-perceived competence (PC), the repeated measure ANOVA showed significant group and measure effects (F = 4.302, *p* < 0.05, η^2^ = 0.137, small; F = 83.002, *p* < 0.001, η^2^ = 0.605, large, respectively). The interaction group × measure was significant too (F = 15.165, *p* < 0.001, η^2^ = 0.349, medium). The post hoc analysis showed higher PC in retest compared to test values in MG and AG (*p* < 0.001 for both comparisons) and higher retest PC perceptions in MG and AG compared to AG (*p* = 0.045) and CG (*p* < 0.001).

In relation to the self-perceived difficulty (PD), the repeated measure ANOVA showed a significant group and measure effects (F = 33.725, *p* < 0.001, η^2^ = 0.555, large; F = 146.935, *p* < 0.001, η^2^ = 0.731, large, respectively). The interaction group × measure was significant too (F = 21.782, *p* < 0.001, η^2^ = 0.446, medium). The post hoc analysis showed lower PD perceptions in retest compared to test values in MG and AG (*p* < 0.001 for both comparisons) and lower retest PD in MG and AG compared to CG (*p* < 0.001 for both comparisons) (Table 3).

## 4. Discussion

The present study aimed to investigate the effect of structure-based program interventions, using combined motor accuracy exercises and visual art activities, on fine motor coordination, selective attention, and reaction time skills in typically developing males of 7–9 years old. The main finding was that the combined program based on motor accuracy exercises and visual art activities best assists accuracy, manual dexterity, reaction time, and selective attention. The childhood period was demonstrated to be golden for enhancing motor competency [1] and cognitive development [16], and is assumed to be of great importance in academic achievement, mental health, and social integration [41]. Nonetheless, the effect of PA on cognitive performance was set to be different according to the type and the environment in which the motor task is undertaken. Indeed, in a review, Shi and Feng [16] highlighted that PA conducted in natural environments has a positive effect on typically developed children and adolescents, and that open and/or sequential features of motor skills are more beneficial in enhancing executive function. Likewise, Wang et al. [42] and Krenn et al. [20] demonstrated that open motor skills compared to closed ones are more beneficial in the improvement of task general cognitive demands. According to Tomporowski and Pesce [43], increasing the complexity and variability of environmental information in the final stages of the motor learning process provides a continuous stimulus to cognition, which in turn largely enhances cognitive benefits. 

The present study findings are in accordance with previous studies reporting that target size and/or distance to target variations better enhance motor accuracy in throwing darts, reduce difficulty, and improve competence task perceptions in children [26,44]. Moreover, Gaspar et al. [27] demonstrated that opting for a differential approach intervention, based on the variation of modalities, is more beneficial in ball kicking accuracy in young football players. The proposed accuracy and mixed programs in the present study included games, motor accuracy, and interception exercises that are initially and principally oriented to enhance motor accuracy. Results portrayed that the aforementioned physical activities (open-ended, strategic, sequential in nature, and with difficulty variations) have affected motor accuracy, manual dexterity, reaction time, selective attention, and executive function [16,43]. While this study did not directly measure executive function, we can assume that the proposed structure-based programs affected it. Indeed, recent research [34] reported that GPT can be used to assess executive function as relationships were proven with the trial-making test (TMT, part B), assessing essentially the executive function, with correlations set at 0.634 and 0.631 for the dominant and the non-dominant hand, respectively. Moreover, games and physical activities based on interaction with peers were found to improve executive function (the abilities of planning and strategy finding) in children [45].

The higher performances in accuracy, manual dexterity, and reaction time recorded in the combined group, in comparison to the accuracy group, can be explained by the contribution of the visual art activities intervention, being effective in visual perception and attention enhancement in children [46]. When implemented in elementary schools, they do indeed provide great benefits to students and have positive effects on their academic achievements [47]. According to Bradeško and Potočnik [25], art-based interventions using various fields of visual art (techniques, materials, expressive methods, and concepts) had a positive effect on the student’s perseverance, attention, and concentration. Hamza and Albanna [48] concluded that a ceramic formation program enhanced fine motor skills in children with Down’s syndrome. Likewise, Tiara et al. [49] highlighted the benefits of finger painting in children with autism as it represents an alternative way to develop their cognitive and fine motor skills through the process of mixing colors and finger manipulations. Moreover, finger painting was shown to be an effective method for increasing focus and attention in children [50]. Thus, in view of the present study findings and the above considerations, the combined accuracy exercises and visual art activities induce amelioration in fine motor coordination and cognitive abilities in children 7–9 years old. Recently, it was stated that “*exercising body and mind is common to physical activity, music, and performance arts, which are all linked with increased cognitive function*” [51]. 

Nonetheless, the present study failed to prove any differences in improvement between the combined and the accuracy groups in selective attention (ΔBT). The result can be explained by the relatively short duration of the structure-based intervention programs (three one-hour weekly sessions), which might be insufficient to produce a substantial difference between the two experimental groups on this cognitive skill. Nonetheless, previous studies [21] demonstrated a significant effect of using art activities (i.e., pottery) in improving preschool children’s creativity and their motor-cognitive skills using a similar program duration (i.e., 16 sessions, each lasting 45 min).

The improvement in fine motor coordination skills, selective attention, and reaction time in both the combined and accuracy groups was accompanied by a concomitant increase in competence and a decrease in difficulty perceptions, which indicated a better state of readiness and commitment of the participants in the motor learning process. In general, individuals who perceive a task as more difficult develop lower levels of perceived competence over time. Those results are in accordance with previous findings [52] stating that one-dimension difficulty manipulation leads to higher competence, lower difficulty perception, and more sustainable fine motor coordination skill learning.

### Strength, Limitations, and Perspectives

To the authors’ best knowledge, the present study is the first to focus on searching for innovative structured interventions aiming at the optimization of fine motor coordination, selective attention, and reaction time in male schoolchildren through the combination of oriented motor accuracy exercises and visual art activities. Notwithstanding the finding of the present study, some limitations merit discussion. First, the structure-based intervention programs lasted two months with three one-hour weekly sessions. Thus, extending the experimentation to a longer duration may allow for showing differences in selective attention between the two experimental groups. Second, cognitive skills like working memory, cognitive flexibility, or executive function have not been measured throughout the investigation. Thus, future investigations should be oriented to elucidate the effect of the combined structure-based program on those cognitive skills. Nonetheless, the present study findings highlight the need for unifying the objectives in school subjects (physical education and art education) and the activation of extra-curricular school activities. Likewise, the findings give a possible new perspective for the physical education curriculum reform [16]. 

## 5. Conclusions

The present study concludes that the proposed combined structured program, based on motor accuracy exercises and visual art activities, best assists fine motor coordination, manual dexterity, and reaction time compared to only the accuracy exercises program in typically developed male schoolchildren of 7–9 years of age. The motor accuracy intervention includes games, competitions, and throwing exercises (darts, beanbags, tennis balls, and table tennis balls) with difficulty manipulation (different target throw sizes and from different distances). However, the visual art activities included ceramics, pottery, and creations using raw materials. Considering the time allocated to both programs (3 sessions of 60 min per week, during 8 weeks), it turns out that altering the task nature (PA, visual art activities) and the modality of execution (difficulty manipulation) best promotes the development of fine motor coordination, manual dexterity, selective attention, and reaction time. Therefore, instructors in sports academies and teachers in schools are advised to use the combined structure-based program in training and in the non-curricular sessions, respectively, to better enhance fine motor coordination and cognitive skills in middle childhood.

## Figures and Tables

**Figure 1 children-10-00786-f001:**
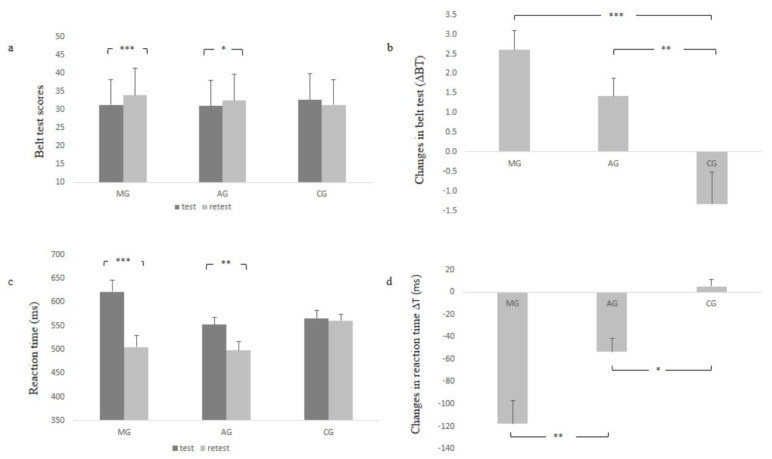
Test-retest performances and changes in selective attention and reaction time performances: (**a**) belt test (BT); (**b**) ΔBT; (**c**) reaction time; (**d**) ΔRT. MG: mixed group; AG: accuracy group; CG: control group. * *p* < 0.05; ** *p* < 0.01; *** *p* < 0.001.

**Figure 2 children-10-00786-f002:**
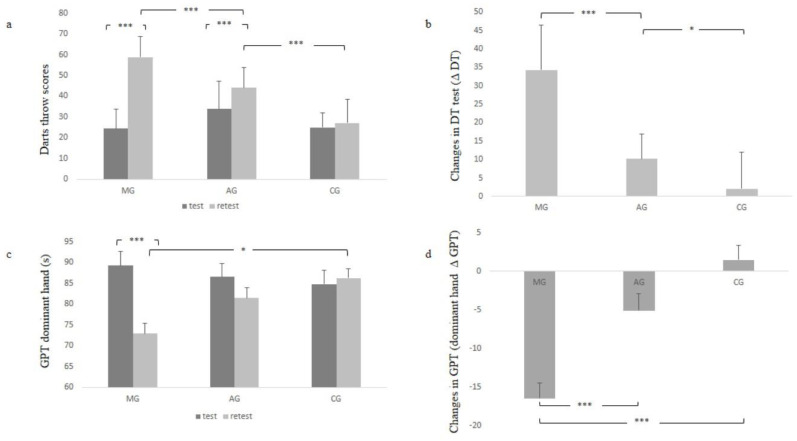
Test-retest performances and changes in darts throw (DT; (**a**,**b**)) and the grooved pegboard tests (GPT; (**c**,**d**)). MG: mixed group; AG: accuracy group; CG: control group. * *p* < 0.05; *** *p* < 0.001.

**Table 1 children-10-00786-t001:** Motor accuracy program exercises.

Activities	Exercises	Instructions	Repetitions-Duration/Session
Games	Battleships game with bean bags	Varying instructions after each game repetition	15 min
	Bean bags toss game		15 min
	JOYIN 12 Wooden Throwing Game		15 min
	Bean bags throws into 3 different-sized buckets and from 3 different distances		9 times
	Softball overhead throws into different-sized buckets, from different distances		9 times
	Catching successively and rapidly thrown tennis balls		>20 times
Darts throws	Darts thrown at different target sizes	Varying difficulties (distance, target size)	>20 times
	Darts throw from different distances		>20 times
	Altering darts thrown from different distances and at different target sizes		>20 times
Tennis ball manipulation and throws	Bouncing a tennis ball with the RH/LH	Varying instructions, altering executions: RH, LH	10 times each
	RH and LH bounce and catch a tennis ball		10 times each
	Bouncing and catching a tennis ball RH/LH		10 times each
	Bouncing a tennis ball from the LH to the LH		10 times
	Alternate Hand Wall Toss		>20 times
	Throwing a ball high and catching it (RH/LH)		>20 times
	Throw and catch a ball with two hands		>20 times
	Throw with one hand and catch with the other hand		>20 times
	Throw and catch a ball to a wall corner		>20 times
	Catching successively and rapidly thrown balls		>20 times
Table tennis ball manipulation and throws	Bouncing a ball up on one side of the racket	Varying difficulties (distance, size)	>20 times
	Bouncing a ball up from one side to the opposite side of the racket		>20 times
	Throwing a ball at different-sized targets and from different distances		>20 times
	Hitting a ball against the wall with a racket		>20 times

RH: right hand; LH: left hand.

**Table 2 children-10-00786-t002:** Visual art activities program.

Field	Art Activities/Instructions	Tools	Sessions/ Duration (min)
Ceramic, pottery formation	Molding: shaping different animals using molds and pressing technique	Molds, clay paste, clay sculpting tools	1 (35 min)
	Free modeling: modeling a car, or a train using colored clay	Colored clay, plasticine, clay sculpting tools	2 (70 min)
	Slice molding, clay bird’s nest molding, and scratching and pressing technique	Clay dough, colorful feathers, colorful beads	2 (70 min)
	Compression molding: shaping a bird out of porcelain or clay	Clay dough, colored feathers, clay sculpting tools	2 (70 min)
	Forming by mold: shaping a fish with pottery and clay	Clay, clay paste, molds, and plastic tools	1 (35 min)
	Shaping using the ropes method: making a vessel by forming clay and pressing in it.	White clay paste, ceramic modeling tools, paints, paintbrushes	2 (70 min)
	Modeling with clay: shaping a snail, a worm, or a duck swimming in a lake	Clay dough in different colors, sculpting tools	1 (35 min)
Creations using raw materials	Formation of environmental and expendable raw materials: forming a plane from wooden sticks	Large and small wooden sticks, small clothespins, glue, colors	2 (70 min
	Shaping with ropes: the child makes vessels by forming the ropes	Rope, adhesive tape, scissors	1 (35 min)
	Forming with slices: making a 3D shape from scraps of paper. Scene of an ant in a field	Colored papers, googly eyes, crayons, different scissors, glue	1 (35 min)
	Forming with slices: making a 3D shape from scraps of paper. Scene of a moving frog	Colored papers, googly eyes, crayons, different scissors, glue	1 (35 min)
	Forming with slices: forming a turtle from paper and knotted strings	Green paper, woolen thread, scissors. Thread winding stabilizer	2 (70 min
	Shaping: creating a mosaic of scraps and colored fabrics	Colored tissue paper or colored medical cork, glue, scissors, pens	1 (35 min)
	Bead arrangement: forming lines of regular beads	Colorful beads, small/medium-sized wooden shapes	2 (70 min)
	Shaping jellyfish from leaves and synthetic clay	Colored papers, colored crafting clay, scissors	1 (35 min)
	Shaping with environmental and consumable materials: shaping a fish	Date kernel, palm leaf, glue	1 (35 min)

**Table 3 children-10-00786-t003:** Test-retest changes in perceived competence and perceived difficulty according to the GPT and DT tests (mean ± SD).

	Groups	Perceived Competence		Perceived Difficulty	
		Test	Retest	Test	Retest
GPT	MG	2.80 ± 0.77	5.20 ± 1.40 ***	9.35 ± 1.57	4.15 ± 1.76 ***
	AG	3.89 ± 2.13	5.11 ± 1.37 **	10.53 ± 2.09	5.37 ± 2.75 ***
	CG	2.82 ± 0.92	3.50 ± 0.86 ^###^	9.35 ± 1.58	9.56 ± 1.65 ^###^
DT	MG	2.75 ± 1.37	5.85 ± 0.98 ***	9.55 ± 2.16	2.90 ± 1.29 ***
	AG	3.21 ± 1.08	4.73 ± 1.28 ***	8.21 ± 2.80	3.31 ± 1.66 ***
	CG	3.11 ± 1.03	3.72 ± 0.91 ^###^	10.55 ± 2.79	9.44 ± 1.88 ^###^

GPT: grooved pegboard test; DT: darts throw test; MG: mixed group; AG: accuracy group; CG: control group; * significantly different from test values in the same group at: ** *p* < 0.01, *** *p* < 0.001; ^#^ significantly different from MG in the same point of measure at: ^###^
*p* < 0.001.

## Data Availability

All datasets used and/or analyses during the current study are available from the corresponding author upon reasonable request.

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
