# Peer review of "Optimizing Fine Motor Coordination, Selective Attention and Reaction Time in Children: Effect of Combined Accuracy Exercises and Visual Art Activities"

_children, 2023, doi:10.3390/children10050786_

Round 1
Reviewer 1 Report
How were these criteria determined? Selection criteria should be stated:
Why were these visual art activities (ceramics, pottery, and raw materials formations) chosen?
Is this time enough? Must explain "two hours of physical education sessions per week."
Discussion; in the first paragraph, I suggest you specify the age range, because it is selective. 320. line: Woss et al. did not provide any age-related findings. It should be checked.
Shi et al. (References; 16, Line 321) meta-analysis. The author should focus on Experimental study results.
337. line: Spearman, no need to test; write correlation.
What do the authors attribute their results to? This issue should be clarified.
In addition, the corrections I added in the explanations in the study should be reviewed again
In the study, I observed that the quality of English was moderate. Its English can be revised by the authors.
Author Response
All co-authors of the manuscript (IDchildren-2351151) entitled “Optimizing Fine Motor Coordination and Cognitive Abilities in Children: Effect of Combined Accuracy Exercises and Visual Art Activities”, want to thank you for your consideration of our work. We agree with the comments advanced by the reviewers and believe that the requested rectifications can improve the quality of the manuscript.
Here you find the point-by-point comments and answers.
All changes in the manuscript in relation to the reviewers’ comments are in red.
Answers to the first reviewer’s questions:
Reviewer one |
||
|
question |
answer |
1 |
How were these criteria determined? Selection criteria should be stated |
Inclusion criteria were added in the method section (L 93-95) The inclusion criteria were: (i) being affiliated with the academy; (ii) not suffering from any intellectual or physical disabilities; (iii) having a rate of attendance in the program > 75%.(iiii) completing all testing sessions. |
2 |
Why were these visual art activities (ceramics, pottery, and raw materials formations) chosen? |
Visual art activities were shown to affect creativity and motor-cognitive skills in both typically developed and handicapped children. All the chosen visual art activities were motivating for children and required reflection, accuracy, and eye-hand coordination. |
3 |
Is this time enough? Must explain "two hours of physical education sessions per week." |
Membership affiliation in the sports academies permits the participants to three one-hour weekly sessions. We agree with the reviewer that this duration may be insufficient to improve the children's motor skills. Hence, we extended the structured intervention program for two months. The structure-based program included 24 training sessions at a rate of three sessions per week and 60 min per session. As participants were PE students, they had two PE classroom sessions during the school time-tale. Nonetheless, a discussion paragraph (L370-377) and study limitations were added (L 391-397). |
4 |
Discussion; in the first paragraph, I suggest you specify the age range, because it is selective. 320. line: Woss et al. did not provide any age-related findings. It should be checked. |
The age range is specified in discussion paragraph 1. The reference Voss et al deleted and replaced by other references. (L325-328; L330-333; 365-369) |
5 |
Shi et al. (References; 16, Line 321) meta-analysis. The author should focus on Experimental study results. |
Additional references were added (experimental studies in particular). |
6 |
337. line: Spearman, no need to test; write correlation. |
Corrected. |
7 |
What do the authors attribute their results to? This issue should be clarified. |
Discussion of the study results was improved, and additional references were added. |
8 |
In the study, I observed that the quality of English was moderate. Its English can be revised by the authors. |
English was revised and all manuscript was language edited. |
I hope that these corrections reach your satisfaction
Thank You for the consideration

Reviewer 2 Report
The topic of the paper is of interest in the field. I am glad to see that visual art can be included in the training of young children successfully. Furthermore, I appreciated a lot that the authors shared the structured program in a table. Overall, I liked the study and with some minor adjustment.
Minor issues:
1. I miss a more accurate description on the participants, especially on the gender distribution of each group. Gender can influence the motor skills at this age; therefore, it is important to see the percentage of boys and girls in the whole and sub-groups. Furthermore, it would be useful to report on age in the 3 subgroups as well.
2. How the sample size was calculated, for the whole sample or for the sub-groups. Please clarify it.
3. The tests used pre and post in the study do not give information on the general cognitive ability but rather on motor skills, dexterity and speed. Maybe Belt test reports on a simple visual attention but not on cognitive ability per se. Therefore authors should be more precise how to use cognitive skills in the introduction, discussion, the abstract, because in this version it seems that cognitive ability is interchangeable of creativity, motor skills, and coping skills. I would think about to change the title as well. Otherwise, the content will not meet readers’ expectation.
The paper seems to me well written, but a native speaker could jugde it better.
Author Response
All co-authors of the manuscript (IDchildren-2351151) entitled “Optimizing Fine Motor Coordination and Cognitive Abilities in Children: Effect of Combined Accuracy Exercises and Visual Art Activities”, want to thank you for your consideration of our work. We agree with the comments advanced by the reviewers and believe that the requested rectifications can improve the quality of the manuscript.
Here you find the point-by-point comments and answers.
All changes in the manuscript in relation to the reviewers’ comments are in red.
Answers to the second reviewer’s questions
Reviewer two |
||
|
question |
answer |
1 |
I miss a more accurate description on the participants, especially on the gender distribution of each group. Gender can influence the motor skills at this age; therefore, it is important to see the percentage of boys and girls in the whole and sub-groups. Furthermore, it would be useful to report on age in the 3 subgroups as well. |
Participants were all schoolchildren (males), having 2 PE sessions per week in school (according to the school timetable). They were affiliated with sports academies and received three one-hour extra-curricular sessions per week. |
2 |
How the sample size was calculated, for the whole sample or for the sub-groups. Please clarify it. |
The a priori sample size was determined using G*Power software (3.1.9.2), given an effect size = 0.25 (Medium), α = 0.05, power 1-β = 0.95, and was set at 57 participants. (L87-88). |
3 |
The tests used pre and post in the study do not give information on the general cognitive ability but rather on motor skills, dexterity and speed. Maybe Belt test reports on a simple visual attention but not on cognitive ability per se. Therefore authors should be more precise how to use cognitive skills in the introduction, discussion, the abstract, because in this version it seems that cognitive ability is interchangeable of creativity, motor skills, and coping skills. I would think about to change the title as well. Otherwise, the content will not meet readers’ expectation. |
We found this remark very relevant as selective attention and reaction time do not cover all cognitive abilities. However we referred to previous studies (Jakobsen et al 2011; Stevens & Bavelier 2012) revealing the importance of reaction time and selective attention in the cognitive functioning in healthy subjects. Nonetheless, a study limitation and perspectives were added at the end of the discussion section.(L 389-395). To be more precise, changes in the title were made as well as in the aim of the study, and discussion, and conclusion.
|
I hope that these corrections reach your satisfaction
Thank You for the consideration
